# Off-Policy Policy Gradient with State Distribution Correction

Yao Liu [1]   Adith Swaminathan [2]   Alekh Agarwal [2]   Emma Brunskill [1]

## Abstract

We study the problem of off-policy policy optimization in Markov decision processes, and develop a novel off-policy policy gradient method. Prior off-policy policy gradient approaches have generally ignored the mismatch between the distribution of states visited under the behavior policy used to collect data, and what would be the distribution of states under the learned policy. Here we build on recent progress for estimating the ratio of the Markov chain stationary distribution of states in policy evaluation, and present an off-policy policy gradient optimization technique that can account for this mismatch in distributions. We present an illustrative example of why this is important, theoretical convergence guarantee for our approach and empirical simulations that highlight the benefits of correcting this mismatch.

## 1. Introduction

The ability to use data about prior decisions and their outcomes to make counterfactual inferences about how alternative decision policies might perform, is a cornerstone of intelligent behavior. It also has immense practical potential – it can enable the use of electronic medical record data to infer better treatment decisions for patients, the use of prior product recommendations to inform more effective strategies for presenting recommendations, and previously collected data from students using educational software to better teach those and future students. Such counterfactual reasoning, particularly when one is deriving decision policies that will be used to make not one but a sequence of decisions, is important since online sampling during a learning procedure is both costly and dangerous, and not practical in many of the applications above. While amply motivated, doing such counterfactual reasoning is also challenging because the data is censored – we can only observe

[1]Department of Computer Science, Stanford University [2]Microsoft Research. Correspondence to: Yao Liu <yaoliu@stanford.edu>.

the result of providing a particular chemotherapy treatment policy to a particular patient, not the counterfactual of if we were then to start with a radiation sequence.

We focus on the problem of performing such counterfactual inferences in the context of sequential decision making in a Markov decision process (MDP). We assume that data has been previously collected using some fixed and known behavior policy, and our goal is to learn a new decision policy with good performance for future use. This problem is often known as batch off-policy policy optimization. We assume that the behavior policy used to gather the data is stochastic: otherwise we will not be able to estimate the performance of any other policy without additional assumptions.

In this paper we consider how to perform batch off-policy policy optimization (OPPO) using a policy gradient method. While there has been increasing interest in batch off-policy reinforcement learning (RL) over the last few years (Thomas et al., 2015; Jiang and Li, 2016; Thomas and Brunskill, 2016), much of this has focused on off-policy policy evaluation, where the goal is to estimate the performance of a particular given target decision policy. Ultimately we will very frequently be interested in the optimization question, which requires us to determine a good new policy for future potential deployment, given a fixed batch of prior data.

To do batch off-policy policy optimization, model free methods (like deep Q-learning (Mnih et al., 2015) or fitted Q iteration (Ernst et al., 2005)) can be used alone, but there are many cases where we might prefer to focus on policy gradient or actor-critic methods. Policy gradient methods have seen substantial success in the last few years (Schulman et al., 2015) in the on-policy setting, and they can be particularly appealing for cases where it is easier to encode inductive bias in the policy space, or when the actions are continuous (see e.g. Abbeel and Schulman (2016) for more discussion). However, existing approaches to incorporating offline information into online policy gradients have shown limited benefit (Gu et al., 2017b;a), in part due to the variance in gradients incurred due to incorporating off-policy data. One approach is to correct exactly for the difference between the sampling data distribution and the target policy data distribution, by using importance sampling to re-weight every sample according to the likelihood ratio of behavior policy and evaluation policy up to that step. Unfortunately

the variance of this importance sampling ratio will grow exponentially with the problem horizon.

To avoid introducing variance in gradients, off-policy actor critic (Off-PAC) (Degris et al., 2012) ignores the stationary state distribution difference between the behavior policy and the target policy, and instead only uses a one step of importance sampling to reweight the action distributions. Many practical off-policy policy algorithms including DDPG (Silver et al., 2014), ACER (Wang et al., 2016), and Off-PAC with emphatic weightings (Imani et al., 2018) are based on the gradient expression in the Off-PAC algorithm (Degris et al., 2012). However as we will demonstrate, not correcting for this mismatch in state distributions can result in poor performance in general, both in theory and empirically.

Instead, here we introduce an off-policy policy gradient algorithm that can be used with batch data and that accounts for the difference in the state distributions between the current target and behavior policies during each gradient step. Our approach builds on recent approaches for policy evaluation that avoid the exponential blow up in importance sampling weights by instead computing a direct ratio over the stationary distribution of state visitations under the target and behavior policy (Hallak and Mannor, 2017; Liu et al., 2018a; Gelada and Bellemare, 2019). We incorporate these ideas within an off-policy actor critic method to do batch policy optimization. We first provide an illustrative example to demonstrate the benefit of this approach over Off-PAC (Degris et al., 2012), and show that correcting for the mismatch in state distributions of the behavior policy and the target policy can be critical for getting good estimates of the policy gradient, and we also provide convergence guarantees for our algorithm under certain assumptions. We then compare our approach and Off-PAC experimentally on two simulated domains, cart pole and a HIV patient simulator (Ernst et al., 2005). Our results show that our approach is able to learn a substantially higher performing policy than both Off-PAC and the behavior policy that is used to gather the batch data. We further demonstrate that we can use the recently proposed off-policy evaluation technique of Liu et al. (2018a) to reliably identify good policies found during the policy gradient optimization run. Our results suggest that directly accounting for the state distribution mismatch can be done without prohibitively increasing the variance during policy gradient evaluations, and that doing so can yield significantly better policies. These results are promising for enabling us to learn better policies given batch data or improving the sample efficiency of online policy gradient methods by being able to better incorporate past data.

**Related Work** Many prior works focus on the policy evaluation problem, as it is a foundation for downstream policy learning problems. These approaches often build on importance sampling techniques to correct for distribution mis-

match in the trajectory space, pioneered by the early work on eligibility traces (Precup et al., 2000), and further enhanced with a variety of variance reduction techniques (Thomas et al., 2015; Jiang and Li, 2016; Thomas and Brunskill, 2016). Some authors consider model-based approaches to OPPE (Farajtabar et al., 2018; Liu et al., 2018b), which usually perform better than importance sampling approaches empirically in policy evaluation settings. But those methods do not extend easily to our OPPO setting, as well as introduce additional challenges due to bias in the models and typically require fitting a separate model for each target policy. The recent work of Liu et al. (2018a) partially alleviates the variance problem for model-free OPPE by reweighting the state visitation distributions, which can result in as just as high a variance in the worst case, but is often much smaller. Our work incorporates this recent estimator in policy optimization methods to enable learning from off-policy collected data.

In the off-policy policy optimization setting, many works study value-function based approaches (like fitted Q iteration (Ernst et al., 2005) and DQN (Mnih et al., 2015)), as they are known to be more robust to distribution mismatch. Some recent works aim to further incorporate reweighting techniques within off-policy value function learning (Hallak and Mannor, 2017; Gelada and Bellemare, 2019). These methods hint at the intriguing potential of value-function based techniques for off-policy learning, and we are interested in similarly understanding the viability of using direct policy optimization techniques in the off-policy setting.

Off-policy actor critic (Degris et al., 2012; Imani et al., 2018) proposed an answer to this question by learning the critic in an off-policy way and reweighting actor gradients by correcting the conditional action probabilities, but ignores the mismatch between the state distributions of the data collection policy and learned policies. A different research thread on trust region policy optimization method (Schulman et al., 2015), while requiring the on-policy setting, incorporates robustness to the mismatch between the data collection and gradient evaluation policies. However this is not a fully off-policy scenario and learning from an offline dataset is still strongly motivated by many applications. Many recent methods (Silver et al., 2014; Wang et al., 2016; Gu et al., 2017a;b; Lillicrap et al., 2015) are derived based on the policy gradient form in Degris et al. (2012) to improve the empirical sample efficiency by using more off-policy samples from previous iteration. In this work, we demonstrate a basic weakness of the policy gradient definition in Degris et al. (2012), and show how to correct that.

## 2. Preliminaries

We consider finite horizon MDPs $M = \langle \mathcal{S}, \mathcal{A}, P, r, \gamma \rangle$, with a continuous state space $\mathcal{S}$, a discrete action space $\mathcal{A}$, a tran-

sition probability distribution $P : \mathcal{S} \times \mathcal{A} \times \mathcal{S} \mapsto [0, 1]$ and an expected reward function $r : \mathcal{S} \times \mathcal{A} \mapsto [0, 1]$. We observe tuples of state, action, reward and next state: $(s_t, a_t, r_t, s_{t+1})$, where $s_0$ is drawn from a initial state distribution $p_0(s)$, action $a$ is drawn from a stochastic behavior policy $\mu(a|s)$ and the reward and next state are generated by the MDP. Given a discount factor $\gamma \in (0, 1]$, the goal is to maximize the expected return of policy:

$$R_M^\pi = \mathbb{E}_\pi \left[ \lim_{T \to \infty} \frac{1}{\sum_{t=0}^T \gamma^t} \sum_{t=0}^T \gamma^t r_t \right] \quad (1)$$

When $\gamma = 1$ this becomes the average reward case and $\gamma < 1$ is called the discounted reward case. Given any fixed policy $\pi$ the MDP becomes a Markov chain and we can define the state distribution at time step $t$: $d_t^\pi(s)$, and the stationary state distribution across time: $d^\pi(s) = \lim_{T \to \infty} \frac{1}{\sum_{t=0}^T \gamma^t} \sum_{t=0}^T \gamma^t d_t^\pi(s)$ To make sure the optimal policy is learnable from collected data, we assume the following about the support set of behavior policy:

**Assumption 1.** *For at least one optimal policy $\pi^*$, $d^\mu(s) > 0$ for all $s$ such that $d^{\pi^*}(s) > 0$, and $\mu(a|s) > 0$ for all $a$ such that $\pi^*(a|s) > 0$ when $d^{\pi^*}(s) > 0$.*

# 3. An Off-Policy Policy Gradient Estimator

Note that Assumption 1 is quite weak when designing a policy evaluation or optimization scheme, since it only guarantees that $\mu$ adequately visits all the states and actions visited by some $\pi^*$. However, a policy optimization algorithm might require off-policy policy gradient estimates at arbitrary intermediate policy it produces along the way, which might visit states not reached by $\mu$. A strong assumption to handle such scenarios is that Assumption 1 holds not just for some $\pi^*$, but any possible policy $\pi$. Instead of making such a strong assumption, we start by defining an augmented MDP where Assumption 1 suffices for obtaining pessimistic estimates of policy values and gradients.

## 3.1. Constructing an Augmented MDP

Given a data collection policy $\mu$, let its support set be $\mathcal{S}_\mu = \{s : d^\mu(s) > 0\}$ and $\mathcal{S}\mathcal{A}_\mu = \{(s,a) : d^\mu(s)\mu(a|s) > 0\}$. Consider a modified MDP $M_\mu = \langle \mathcal{S}_\mu \bigcup \{s_{abs}\}, \mathcal{A}, P_\mu, r_\mu, \gamma \rangle$. Any state-action pairs not in $\mathcal{S}\mathcal{A}_\mu$ will essentially transition to $s_{abs}$ which is a new absorbing state where all actions will lead to a zero reward self-loop. Concretely, $P_\mu(s_{abs}|s_{abs}, a) = 1$ and $r(s_{abs}, a) = 0$ for any $a$. For all other states, the transition probabilities and rewards are defined as: For $(s, a) \in \mathcal{S}\mathcal{A}_\mu$, $P_\mu(s'|s, a) = P(s'|s, a)$ for all $s' \in \mathcal{S}_\mu$, and $P_\mu(s_{abs}|s, a) = \int_{s \notin \mathcal{S}_\mu} P(s'|s, a)ds'$. For all $s \in \mathcal{S}_\mu$ but $(s, a) \notin \mathcal{S}\mathcal{A}_\mu$, $P_\mu(s_{abs}|s, a) = 1$. $r_\mu(s, a) = r(s, a)$ for

$(s, a) \in \mathcal{S}\mathcal{A}_\mu$, and $r_\mu(s, a) = 0$ otherwise. First we prove that the optimal policy $\pi^*$ of the original MDP remains optimal in augmented MDP as a consequence of Assumption 1.

**Theorem 1.** *The expected return of all policies $\pi$ in the original MDP is larger than the expected return in the new MDP: $R_M^\pi \geq R_{M_\mu}^\pi$. For any optimal $\pi^*$ that satisfies Assumption 1 we have that $R_M^{\pi^*} = R_{M_\mu}^{\pi^*}$*

That is, policy optimization in $M_\mu$ has at least one optimal solution identical to the original MDP $M$ with the same policy value since $M_\mu$ lower bounds the policy value in $M$, so sub-optimal policies remain sub-optimal.

*Proof.* For any trajectory sampled from policy $\pi$, if every $s_k, a_k \in \mathcal{S}\mathcal{A}_\mu$ then $\sum_{t=0}^T \gamma^t r(s_t, a_t) = \sum_{t=0}^T \gamma^t r_\mu(s_t, a_t)$. If not, let $s_{k+1}, a_{k+1}$ be the first state-action pair that is not in $\mathcal{S}\mathcal{A}_\mu$. Then $\sum_{t=0}^T \gamma^t r(s_t, a_t) \geq \sum_{t=0}^k \gamma^t r(s_t, a_t) = \sum_{t=0}^k \gamma^t r_\mu(s_t, a_t) + \sum_{t=k+1}^k \gamma^t r_\mu(s_{abs}, a_t)$. Dividing the accumulated rewards by $\frac{1}{\sum_{t=0}^T \gamma^t}$ and taking the limit of $T \to \infty$, then taking the expectation over trajectories induced by $\pi$, we have that: $R_M^\pi \geq R_{M_\mu}^\pi$. For $\pi^*$, since $\mathcal{S}\mathcal{A}_\mu$ covers all state-action pairs reachable by $\pi^*$, so the expected return remains the same. □

## 3.2. Off-Policy Policy Gradient in Augmented MDP

We will now use the expected return in the modified MDP, $R_{M_\mu}^\pi$, as a surrogate for deriving policy gradients. According to the policy gradient theorem in Sutton et al. (2000), for a parametric policy $\pi$ with parameters $\theta$:

$$\frac{\partial R_{M_\mu}^\pi}{\partial \theta} = \sum_s d^\pi(s) \sum_a \pi(a|s) \frac{\partial \log \pi(a|s)}{\partial \theta} Q_{M_\mu}^\pi(s, a).$$

From here on, $d^\pi(s)$ is with respect to the new MDP. Now we will show that we can get an unbiased estimator of this gradient using importance sampling from the stationary state distribution $d^\mu(s)$ and the action distribution $\mu(a|s)$. According to the definition of $M_\mu$, we have that for all $s, a$ such that $d^\mu(s)\mu(a|s) = 0$, $(s, a)$ is not in $\mathcal{S}\mathcal{A}_\mu$. Hence $Q_{M_\mu}^\pi(s, a) = 0$ for any policy $\pi$ since $(s, a)$ will receive zero reward and lead to a zero reward self-loop. So we have:

$$\frac{\partial R_{M_\mu}^\pi}{\partial \theta} = \sum_s d^\pi(s) \sum_a \pi(a|s) \frac{\partial \log \pi(a|s)}{\partial \theta} Q_{M_\mu}^\pi(s, a)$$

$$= \sum_{s : d^\mu(s) > 0} \frac{d^\pi(s)}{d^\mu(s)} d^\mu(s)$$

$$\sum_{a : \mu(a|s) > 0} \frac{\pi(a|s)}{\mu(a|s)} \mu(a|s) \frac{\partial \log \pi(a|s)}{\partial \theta} Q_{M_\mu}^\pi(s, a)$$

$$= \mathbb{E}_{d^\mu, \mu} \left[ \frac{d^\pi(s)}{d^\mu(s)} \frac{\pi(a|s)}{\mu(a|s)} \frac{\partial \log \pi(a|s)}{\partial \theta} Q_{M_\mu}^\pi(s, a) \right] \quad (2)$$

Note that according to the definition of $M_\mu$, the Markov chain induced by $M$ and $\mu$ is exactly the same as $M_\mu$ and $\mu$. Thus the distribution of $(s_t, a_t, s_{t+1})$ generated by executing $\mu$ in $M$ is the same as executing $\mu$ in $M_\mu$. So, we can estimate this policy gradient using the data we collected from $\mu$ in $M$. We conclude the section by pointing out that working in the augmented MDP allows us to construct a reasonable off-policy policy gradient estimator under the mild Assumption 1, while all prior works in this vein explicitly or implicitly require the coverage of all possible policies.

Note that in the average reward case, such an augmented MDP would not be helpful for policy optimization since all policies that potentially reach $s_{abs}$ will have a value of zero, and the stationary state distribution will be a single mass in the absorbing state. That would not induce a practical policy optimization algorithm. In the average reward case, either we need a stronger assumption that $\mu$ covers the entire state-action space or we must approximate the problem by setting a discount factor $\gamma < 1$ for the policy optimization algorithm, which is a common approach for deriving practical algorithms in an average reward (episodic) environment.

## 4. Algorithm: OPPOSD

Given the off-policy policy gradient derived in (2), how can we efficiently estimate it from samples collected from $\mu$? Notice that most quantities in the gradient estimator (2) are quite intuitive and also present in prior works such as Off-PAC. The main difference is the state distribution reweighting $d^\pi(s)/d^\mu(s)$, which we would like to estimate using samples collected with $\mu$. For estimating this ratio of state distributions, we build on the recent work of Liu et al. (2018a) which we describe next.

For a policy $\pi$, let us define the shorthand $\rho_\pi(s, a) = \pi(s, a)/\mu(s, a)$. Further given a function $w : \mathcal{S} \to \mathbb{R}$, define $\Delta(w; s, a, s') := w(s)\rho_\pi(s, a) - w(s')$. Then we have the following result.

**Theorem 2** ((Liu et al., 2018a)). *Given any* $\gamma \in (0, 1)$, *assume that* $d^\mu(s) > 0$ *for all* $s$ *and define*

$$L(w, f) = \gamma \mathbb{E}_{(s,a,s')\sim d^\mu}[\Delta(w; s, a, s')f(s')] \\ + (1 - \gamma)\mathbb{E}_{s\sim p_0}[(1 - w(s))f(s)].$$

*Then* $w(s) = d^\pi(s)/d^\mu(s)$ *if and only if* $L(w, f) = 0$ *for any measurable test function* $f$.[1]

This result suggests a constructive procedure for estimating the state distribution ratio using samples from $\mu$, by finding a function $w$ over the states which minimizes $\max_f L(w, f)$. Since the maximization over all measurable functions as per

Theorem 2 is intractable, Liu et al. (2018a) suggest restricting the maximization to a unit ball in an RKHS, which has an analytical solution to the maximization problem, and we use the same procedure to approximate density ratios in our algorithm.

Applying Theorem 2 requires overcoming one final obstacle. The theorem presupposes $d^\mu(s) > 0$ for all $s$. In case where $\mathcal{SA}_\mu = \mathcal{S} \times \mathcal{A}$ we can directly apply the theorem. Otherwise in the MDP $M_\mu$, this assumption indeed holds for all states, but $\mu$ never visits the absorbing state $s_{abs}$, or any transitions leading into this state. However, since we know this special state, as well as the dynamics leading in and out of it, we can simulate some samples for this state, effectively corresponding to a slight perturbation of $\mu$ to cover $s_{abs}$. Concretely, we first choose a small smoothing factor $\epsilon \in (0, 1)$. For any sample $(s, a, s')$ in our data set, if there exist $k$ actions $\widetilde{a}$ such that $\mu(\widetilde{a}|s) = 0$, then we will keep the old samples with probability $1 - \epsilon$ and sample any one of the $k$ actions with probability $\epsilon/k$ uniformly and change the next state $s'$ to $s_{abs}$. If we sampled $\widetilde{a}$, consequently, we would also change all samples after this transition to a self-loop in $s_{abs}$. Thus we create samples drawn according to a new behavior policy which covers all the state action pairs: $\widetilde{\mu} = (1 - \epsilon)\mu + \epsilon U(s)$ where $U(s)$ is a uniform distribution over the $k$ actions not chosen by $\mu$ in state $s$. Now we can use Theorem 2 and the algorithm from Liu et al. (2018a) to estimate $d^\pi(s)/d^{\widetilde{\mu}}(s)$. Note that the propensity scores and policy gradients computed on this new dataset correspond to the behaviour policy $\widetilde{\mu}$ and not $\mu$. Formally, in place of using (2), we now estimate:

$$\mathbb{E}_{d^{\widetilde{\mu}}, \widetilde{\mu}} \left[ \frac{d^\pi(s)}{d^{\widetilde{\mu}}(s)} \frac{\pi(a|s)}{\widetilde{\mu}(a|s)} \frac{\partial \log \pi(a|s)}{\partial \theta} Q^\pi_{M_\mu}(s, a) \right] \quad (3)$$

Note that we can estimate the expectation in (3) from the smoothed dataset by construction, since the ratio $\pi(s, a)/\widetilde{\mu}(s, a)$ in all states are known.

Now that we have an algorithm for estimating policy gradients from (3), we can plug this into any policy gradient optimization method. Following prior work, we incorporate our off-policy gradients into an actor-critic algorithm. For learning the critic $Q^\pi_{M_\mu}(s, a)$, we can use any off-policy Temporal Difference (Bhatnagar et al., 2009; Maei, 2011) or Q-learning algorithm (Watkins and Dayan, 1992). In our algorithm, we fit an approximate value function $\widehat{V}$ by: [2]

$$\widehat{V}(s) \leftarrow \widehat{V}(s) + \alpha_c \frac{\pi(a|s)}{\widetilde{\mu}(a|s)} \left( R^\lambda(s, a) - \widehat{V}(s) \right), \quad (4)$$

where $\alpha_c$ is the step-size for critic updates and $R^\lambda(s, a)$ is

---

[1] When $\gamma = 1$, $w$ is only determined up to normalization, and hence an additional constraint $\mathbb{E}_{s\sim d^\mu}[w(s)] = 1$ is required to obtain the conclusion $w(s) = d^\pi(s)/d^\mu(s)$.

[2] For simplicity, Eqn 4 views $\widehat{V}(s)$ in the tabular setting. See Line 14 in Alg 1 for the function approximation case.

the off-policy $\lambda$-return:

$$R^\lambda(s,a) = r(s,a) + (1-\lambda)\gamma\widehat{V}(s') + \lambda\gamma\frac{\pi(a|s)}{\widetilde{\mu}(a|s)}R^\lambda(s',a'),$$

and $(s,a,s',a')$ is generated by executing $\widetilde{\mu}$. After we learn $\widehat{V}$, $R^\lambda$ serves the role of $Q^\pi_{M_\mu}$ in our algorithm.

Given the estimates of the state distribution ratio from Liu et al. (2018a) and the critic updates from (4), we can now update the policy by plugging these quantities in (3). It remains to specify the initial conditions to start the algorithm. Since we have data collected from a behavior policy, it is natural to also warm-start the actor policy in our algorithm to be the same as the behavior policy and correspondingly the critic and $w$'s to be the value function and distribution ratios for the behavior policy. This can be particularly useful in situations where the behavior policy, while suboptimal, still gets to states with high rewards with a reasonable probability. Hence we use behavior cloning to warm-start the policy parameters for the actor, use on-policy value function learning for the critic and also fit the state ratios $w$ for the actor obtained by behavior cloning. Note that while the ratio will be identically equal to 1 if our behavior cloning was perfect, we actually estimate the ratio to better handle imperfections in the learned actor initialization.

A full pseudo-code of our algorithm, which we call OPPOSD for Off-Policy Policy Optimization with State Distribution Correction, is presented in Algorithm 1 in appendix. We mention a couple of implementation details which we found helpful in improving the convergence properties of the algorithm. Typical actor-critic algorithms update the critic once per actor update in the on-policy setting. However, in the off-policy setting, we find that performing multiple critic updates before an actor update is helpful, since the off-policy TD learning procedure can have a high variance. Secondly, the computation of the state distribution ratio $w$ is done in an online manner similar to the critic updates, and analogous to the critic, we always retain the state of the optimizer for $w$ across the actor updates (rather than learning the $w$ from scratch after each actor update). Similar to the critic, we also perform multiple $w$ updates after each actor update. These choices are intuitively reasonable as the standard two-time scale asymptotic analysis of actor-critic methods (Borkar, 2009) does require the critic to converge faster than the actor.

## 5. Convergence Result

In this section, we present two main results to demonstrate the theoretical advantage of our algorithm. First we present a simple scenario where the prior approach of Off-PAC yields an arbitrarily biased gradient estimate, despite having access to a perfect critic. In contrast OPPOSD estimates the gradients correctly whenever the distribution ratios in (2)

and the critic are estimated perfectly, by definition. We will further provide a convergence result for OPPOSD to a stationary point of the expected reward function.

**A hard example for Off-PAC** Many prior off-policy policy gradient methods use the policy gradient estimates proposed in Degris et al. (2012).

$$g_{\text{OffPAC}}(\theta) = \sum_s d^\mu(s) \sum_a \pi(a|s)\frac{\partial \log \pi(a|s)}{\partial\theta}Q^\pi(s,a)$$

Notice that, in contrast to the exact policy gradient, the expectation over states is taken with respect to the behavior policy $\mu$ distribution instead of $\pi$. In tabular settings this can lead to correct policy updates, as proved by Degris et al. (2012). We now present an example where the policy gradient computed this way is problematic when using function approximators. Consider the problem instance shown in Figure 1, where the behavior policy $\pi_b$ is given as: $\pi_b(s_i, \ell) = 0.5$ for $i = 1, 2, \ldots 8$. Thus $\pi_b$ gives us good coverage over all states and actions. Now we consider policies parameterized by a parameter $\alpha \in [0,1]$ where $\pi_\alpha$ has the following structure: $\pi_\alpha(s_0, \ell) = \pi_\alpha(s_3, \ell) = \pi_\alpha(s_4, \ell) = 1$, $\pi_\alpha(s_1, \ell) = \pi_\alpha(s_2, \ell) = \alpha$. Thus $\pi_\alpha$ aliases the states $s_3$ and $s_4$ as a manifestation of imperfect representation which is typical with large state spaces. The true state value function of $\pi_\alpha$, $V_{\pi_\alpha}$ satisfies that: $V^{\pi_\alpha}(s_0) = V^{\pi_\alpha}(s_1) = \frac{1+\alpha}{2}$, $V^{\pi_\alpha}(s_2) = \frac{1-\alpha}{2}$, $V^{\pi_\alpha}(s_3) = 1$, $V^{\pi_\alpha}(s_4) = 0$. Now we define our policy class $\Pi = \{\pi_\alpha : \alpha \in [0,1]\}$. Clearly the optimal policy is $\pi_1$ as it completely eliminates the ill-effects of state aliasing. We now study the Off-PAC gradient estimator $g_{\text{OffPAC}}(\alpha)$ in an idealized setting where the critic $Q^{\pi_\alpha}$ is perfectly known. As per Equation 5 of Degris et al. (2012), we have

$$\begin{aligned}
g_{\text{OffPAC}}(\alpha) &= \sum_s d^{\pi_b}(s) \sum_a \frac{\partial \pi_\alpha(a|s)}{\partial\alpha}Q^{\pi_\alpha}(s,a) \\
&= d^{\pi_b}(s_1)\Big(Q^{\pi_\alpha}(s_1,\ell) - Q^{\pi_\alpha}(s_1,r)\Big) \\
&\quad + d^{\pi_b}(s_2)\Big(Q^{\pi_\alpha}(s_2,\ell) - Q^{\pi_\alpha}(s_2,r)\Big) \\
&= \tfrac{1}{2}(1 - 1/2) + \tfrac{1}{2}(0 - 1/2) = 0.
\end{aligned}$$

That is, the gradient vanishes for any policy $\pi_\alpha$, meaning that the algorithm can be arbitrarily sub-optimal at any point during policy optimization. We note that this does not contradict the previous Off-PAC theorems as the policy class is not fully expressive in our example, a requirement for their convergence results. Our gradient estimator (2) instead evaluates to $\partial R^{\pi_\alpha}_{M_\mu}/\partial\alpha = (1+\alpha)/4$, which is correctly maximized at $\alpha = 1$.

**Convergence results for OPPOSD** We next ask whether OPPOSD converges, given reasonable estimates for the density ratio and the critic. To this end, we need to introduce

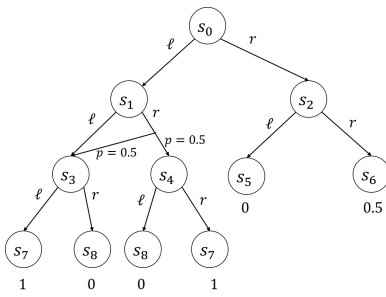

*Figure 1.* Hard example for Off-PAC (Degris et al., 2012)

some additional notations and assumptions. Suppose we run OPPOSD over some parametric policy class $\pi_\theta$ with $\theta \in \Theta$. In the sequel, we use subscripts and superscripts by $\theta$ to mean the corresponding quantities with $\pi_\theta$ to ease the notation. We begin by describing an abstract set of assumptions and a general result about the convergence of OPPOSD, when we run it over the policies $\pi_\theta$ given data collected with an arbitrary policy $\nu$, before instantiating our assumptions for the specific structure of $\widetilde{\mu}$ used in our algorithm.

**Definition 1.** *A function* $f : \mathbb{R}^d \to \mathbb{R}$ *is L-smooth when*

$$\|\nabla f(x) - \nabla f(y)\|_2 \le L\|x - y\|_2, \quad \text{for all } x, y \in \mathbb{R}^d,$$

**Assumption 2.** $\forall (s, a)$ *pairs,* $\forall \theta \in \Theta$ *and a data collection policy* $\nu$, *we assume that the MDP guarantees:*

1. $\left\| \frac{\partial \pi_\theta(a|s)}{\partial \theta} \right\| \le G_{\max}$
2. $Q^\theta(s, a) \le V_{\max}$
3. $\nu(a|s) \ge \nu_{\min}$
4. $w(s) := d^\theta(s)/d^\nu(s) \le w_{\max}$, *and the function approximator* $\hat{w}(s)$ *for* $w(s)$ *satisfies* $\hat{w}(s) \le w_{\max}$.
5. *The expected return of* $\pi_\theta$: $R^\theta$ *is a differentiable, G-Lipschitz and L-smooth function w.r.t.* $\theta$.

**Theorem 3.** *Assume an MDP, a data collection policy* $\nu$ *and function classes* $\{\pi_\theta\}$ *and* $\{\hat{w}\}$ *satisfy Assumption 2. Suppose OPPOSD with policy parameters* $\theta_k$ *at iteration* $k$ *is provided critic estimates* $\widehat{Q}_k$ *and distribution ratio estimates* $\hat{w}_k$ *satisfying* $\mathbb{E}_{(s,a)\sim d^\nu}(w_{\theta_k}(s,a) - \hat{w}_k(s,a))^2 \le \varepsilon_{w,k}^2$ *and* $\mathbb{E}_{(s,a)\sim d^\nu}(Q^{\theta_k}(s,a) - \widehat{Q}_k(s,a))^2 \le \varepsilon_{Q,k}^2$ *for iterations* $k = 1, 2, \ldots K$. *Then*

$$\frac{1}{K}\sum_{k=1}^{K} \mathbb{E}\left[\left\|\nabla_\theta R^{\theta_k}\right\|^2\right] \le \frac{2V_{\max}}{K}$$
$$+ \frac{\sum_{k=1}^{K} O\big((\varepsilon_{w,k}^2 V_{\max}^2 + \varepsilon_{Q,k}^2 w_{\max}^2)G_{\max}^2\big)}{K\nu_{\min}^2}. \quad (5)$$

That is, when Assumption 2 holds, the scheme converges to an approximate stationary point given estimators $\hat{w}$ and $\widehat{Q}$

with a small average MSE across the iterations under $\nu$. An immediate consequence of the theorem above is that as long as we guarantee that $\lim_{K\to\infty} \frac{\sum \varepsilon_{w,k}^2 + \varepsilon_{Q,k}^2}{K} = 0$, which a reasonable online critic and $w$ learning algorithm can guarantee, we have: $\lim_{K\to\infty} \frac{1}{K}\sum_{k=1}^{K} \mathbb{E}\left[\left\|\nabla_\theta R_{M_\nu}^{\theta_k}\right\|^2\right] = 0$, which implies the procedure will converge to a stationary point where the true policy gradient is zero.

We now discuss the validity of Assumption 2 in the specific context of the data collection policy $\widetilde{\mu}$ used in OPPOSD as well as the augmented MDP $M_\mu$. The first assumption, that the gradient of policy distribution is bounded, can be achieved by an appropriate policy parametrization such as a linear or a differentiable neural network-based scoring function composed with a softmax link. The second assumption on bounded value functions is standard in the literature. In particular, both these assumptions are crucial for the convergence of policy gradient methods even in an on-policy setting. The third assumption on lower bounded action probabilities holds by construction for the policy $\widetilde{\mu}$ due to the $\epsilon$−smoothing. The fourth assumption on bounded distribution ratios can be ensured if $d^{\widetilde{\mu}}(s) \ge 1/w_{\max}$. Technically, this might not hold for $\widetilde{\mu}$ in $M_\mu$ as some states in $\mathcal{S}_\mu$ might be reached with tiny probabilities, but we can instead define $\mathcal{S}_\mu$ to be the set of all the states with $d^\mu(s) \ge 1/w_{\max}$. With this change, and given a suitably large $\epsilon$, $\widetilde{\mu}$ always satisfies the fourth assumption in the MDP $M_\mu$. We note that the assumption also requires the outputs of the function approximator $\hat{w}(s)$ to be bounded, which might require additional clipping or regularization in the algorithm. In Algorithm 1, we instead use a weighted importance sampling version of $\hat{w}$ which normalize the value in $\hat{w}$ by its mean in one batch, which ensures that the largest value of $w$ is the mini-batch size $|B_a|$. Finally the regularity assumption on the smoothness of the reward function is again standard for policy gradient methods even in an on-policy setting.

Thus we find that under standard assumptions for policy gradient methods, along with some reasonable additional conditions, we expect OPPOSD to have good convergence properties in theory.

## 6. Experimental Evaluation

In this section we study the empirical properties of OPPOSD, with an eye towards two questions:

1. Does the state distribution correction help improve the performance of off-policy policy optimization?
2. Can we identify the best policy from the optimization path using off-policy policy evaluation?

**Baseline and implementation details** To answer the first question, we compare OPPOSD with its closest prior work,

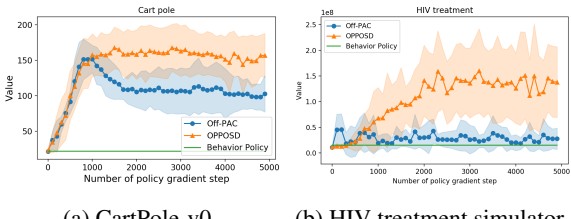

(a) CartPole-v0.  (b) HIV treatment simulator.

*Figure 2.* Episodic scores over length 200 episodes in CartPole-v0 (Barto et al., 1983; Brockman et al., 2016) (left) and HIV treatment simulator (Ernst et al., 2006) (right). Shaded region represents 1 standard deviation over 10 runs of each method.

but without the state distribution correction, that is the Off-PAC algorithm (Degris et al., 2012).

We implement both OPPOSD and Off-PAC using feedforward neural networks for the actor and critic, with ReLU hidden layers. For state distribution ratio $w$, we also use a neural network with ReLU hidden layers, with the last activation function $f(x) = \log(1 + \exp(x))$ to guarantee that $w(s) > 0$ for any input. To make a fair comparison, we keep the implementation of Off-PAC as close as possible to OPPOSD other than the use of $w$. Concretely, we also equip Off-PAC with the enhancements that we find improve empirical performance such as warm start of the actor and critic, as well as several critic updates per actor update. We use the same off-policy critic learning algorithm for Off-PAC and OPPOSD. To learn $w$, we use Algorithm 1 (average reward) in Liu et al. (2018a) with RBF kernel for CartPole-v0 experiment, and Algorithm 2 (discounted reward) in Liu et al. (2018a) with RBF kernel for HIV experiment. We normalize the input to the networks to have 0 mean and 1 standard deviation, and in each mini-batch we normalized kernel loss of fitting $w$ by the mean of the kernel matrix elements, to minimize the effect kernel hyper-parameters on the learning rate. Full implementation details when omitted are provided in the Appendix.

**Simulation domains**  We compare the algorithms in two simulation domains. The first domain is the *cart pole* control problem, where an agent needs to balance a mass attached to a pole in an upright position, by applying one of two sideways movements to a cart on a frictionless track. The state space is continuous and describes the position and velocity of cart and pole. The action space consists of applying a unit force to two directions. The horizon is fixed to 200. If the trajectory ends in less than 200 steps, we pad the episode by continuing to sample actions and repeating the last state. We use a uniformly random policy to collect $n = 500$ trajectories as off-policy data, which is a very challenging data set for off-policy policy optimization methods to learn from as this policy does not attain the desired upright configuration for any prolonged period of

time. We use neural networks with a 32-unit hidden layer to fit the stationary distribution ratio, actor and critic.

The second domain is an *HIV treatment simulation* described in (Ernst et al., 2006). Here the states are six-dimensional real-valued vectors, which model the response of numbers of cells/virus to a treatment. Each action corresponds to whether or not to apply two types of drug, leading to a total of 4 actions. The transition dynamics are modeled by an ODE system in Ernst et al. (2006). The reward consists of a small negative reward for deploying each type of drug, and a positive reward based on the HIV-specific cytotoxic T-cells which will increase with a proper treatment schedule. To maximize the total reward in this simulator, algorithms need to do structured treatment interruption (STI), which aim to achieve a balance between treatment and the adverse effect of abusing drugs. The horizon of this domain is 200 and discount factor is set by the simulator to $\gamma = 0.98$. Each trajectory simulates a treatment period for one patient in 1000 days and each step corresponds to a 5-day interval in the ODE system. We represent the state by taking logarithm of state features and divide the reward by $10^8$ to ensure they are in a reasonable range to fit the neural network models. A uniformly random policy does not visit any rewarding states often enough to collect useful data for off-policy learning. To simulate an imperfect but reasonable data collection policy, we first train an on-policy actor critic method to learn a reasonable (but still far from optimal) policy $\hat{\pi}$. We then use the data collection policy $\mu = 0.7 * \hat{\pi} + 0.3 * U$, where $U$ is the uniformly random policy, to collect $n = 1000$ trajectories. We use neural networks with three 16-unit hidden layers to fit the actor and state distribution ratio, and a neural network with four 32-unit hidden layers for the critic.

Though in both domains our data collection policy is eventually able to cover the whole state-action space, the situation under finite amounts of data is different. In cart pole since an optimal policy can control the cart to stay in a small region, it is relatively easy for the uniform random policy to cover the states visited by the optimal policy. In the HIV treatment domain, it is unlikely that the logged data will cover the desirable state space.

**Impact of state reweighting on policy optimization**  In Figures 2a and 2b, we plot the on-policy evaluation values of the policies produced by OPPOSD and Off-PAC during the actor updates across 10 runs. Each run uses a different data set collected using the behavior policy as well as a different random seed for the policy optimization procedure. In each run we use the same dataset for each method to allow paired comparisons. We evaluate the policy after every 100 actor updates using on-policy Monte-Carlo evaluation over 20 trajectories. The results are averaged over 10 runs and error bars show the standard deviation. Along X-axis, the

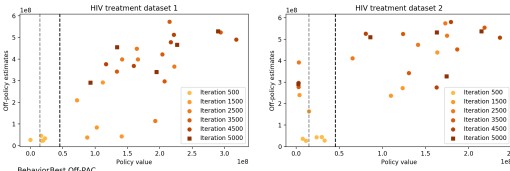

*Figure 3.* Off-policy policy evaluation results of saved policies from OPPOSD. The estimated and true values exhibit a high correlation (coefficient = 0.80 and 0.71 in the left and right plots) for most policies. Two panels correspond to repeating the whole procedure using two datasets from the same behavior policy.

plot shows how the policy value changes as we take policy gradient steps.

At a high-level, we see that in both the domains our algorithm significantly improves upon the behavior policy, and eventually outperforms Off-PAC consistently. Zooming in a bit, we see that for the initial iterates on the left of the plots, the gap between OPPOSD and Off-PAC is small as the state distribution between the learned policies is likely close enough to the behavior policy for the distribution mismatch to not matter significantly. This effect is particularly pronounced in Figure 2a. However, the gap quickly widens as we move to the right in both the figures. In particular, Off-PAC barely improves over behavior policy in Figure 2b, while OPPOSD finds a significantly better policy. Overall, we find that these results are an encouraging validation of our intuition about the importance of correcting the state distribution mismatch.

**Identifying Best Policy by Off-Policy Evaluation**
While OPPOSD consistently outperforms Off-PAC in average performance across 10 runs in both the domains, there is still significant variance in both the methods across runs. Given this variance, a natural question is whether we can identify the best performing policies, during and across multiple runs of OPPOSD for a single dataset. To answer this question, we checkpoint all the policies produced by OPPOSD after every 1000 actor updates, across 5 runs of our algorithm with the same input dataset generated in the HIV domain. We then evaluate these policies using the off-policy policy evaluation (OPPE) method in Liu et al. (2018a). The evaluation is performed with an additional dataset sampled from the behavior policy.

We show the quality of the OPPE estimates against the true policy values for two different datasets for OPPE sampled from the behavior policy in the two panels of Figure 3. In each plot, the X-axis shows the true values by on-policy Monte-Carlo evaluation results and Y-axis shows the OPPE estimates. We find that the OPPE estimates are generally well correlated with the on-policy values, and picking the policy with the best OPPE estimate results in a true value

substantially better than both the best Off-PAC result as well as the behavior policy. A closer inspection also reveals the importance of this validation step. The red squares correspond to the final iterate of OPPOSD in each of the 5 iterations, which has a very high value in some cases, but somewhat worse in other runs. Using OPPE to robustly select a good policy adds a layer of additional assurance to our policy optimization procedure.

# 7. Discussion and Conclusion

We presented a new off-policy actor critic algorithm, OPPOSD, based on a recently proposed stationary state distribution ratio estimator. There exist many interesting next steps, including different critic learning methods which may also leverage the state distribution ratio, and exploring alternative methods for policy evaluation or alternative stationary state distribution ratio estimators (Hallak and Mannor, 2017; Gelada and Bellemare, 2019). Another interesting direction is to improve the sample efficiency of online policy gradient algorithms by using our corrected gradient estimates.

In parallel with our work, Zhang et al. (2019) have presented a different approach for off-policy policy gradient, motivated by a similar recognition of the bias in the Off-PAC gradient estimator. While similarly motivated, the two works have important differences. On the methodological side, Zhang et al. (2019) start from an off-policy objective function and derive a gradient for it. In contrast, we compute an off-policy estimator for the gradient of the on-policy objective function. The latter leads to a much simpler method, both conceptually and computationally, as we do not need to compute the gradients of the visitation distribution. On the other hand, Zhang et al. (2019) focus on incorporating more general interest functions in the off-policy objective, and use the emphatic weighting machinery for obtaining the gradient of their off-policy objective. In terms of settings, our approach works in the offline setting (though easily extended to online), while they require an online setting in order to compute the gradients of the propensity score function. Finally, we present convergence results quantifying the error in our critic and propensity score computations while Zhang et al. (2019) assume a perfect oracle for both and rely on a truly unbiased gradient estimator for the convergence results.

To conclude, our algorithm fixes the bias in off-policy policy gradient estimates introduced by the behavior policy's stationary state distribution. We prove under certain assumptions our algorithm is guaranteed to converge. We also show that ignoring the bias due to the mismatch in state distributions can make an off policy gradient algorithm fail even in a simple illustrative example, and that by accounting for this mismatch our approach yields significantly better performance in two simulation domains.

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

## A. Algorithm Psuedo-code

---

**Algorithm 1** OPPOSD: Off-Policy Policy Optimization with State Distribution Correction

---

**Require:** $\mathcal{S}, \mathcal{A}, \mu, \mathcal{D} : \left\{ \{s_t^i, a_t^i, r_t^i, \mu(a_t^i | s_t^i)\}_{t=0}^T \right\}_{i=0}^n$
**Require:** Hyperparameters $\lambda, \gamma, N_w, \alpha_w, N_c, \alpha_c, \alpha$
1: Warm start $\pi_\theta, \widehat{V}_{\theta_c}, w_{\theta_w}$
2: Pad $\mathcal{D}$ to get $\mathcal{D}', \widetilde{\mu}$ if necessary
3: **for** each step of policy update **do**
4:      **for** state ratio updates $i = 1, 2, \ldots, N_w$ **do**
5:          Sample a mini-batch $B_w \sim \mathcal{D}'$ according to $\widehat{d}_\gamma$ [3]
6:          **if** $\gamma = 1$ **then**
7:              Perform one update according to Algorithm 1 in Liu et al. (2018a) with stepsize $\alpha_w$
8:          **else**
9:              Perform one update according to Algorithm 2 in Liu et al. (2018a) with stepsize $\alpha_w$
10:          **end if**
11:      **end for**
12:      **for** critic updates $i = 1, 2, \ldots, N_c$ **do**
13:          Sample a mini-batch $B_c \sim \mathcal{D}'$
14:          $\theta_c \leftarrow \theta_c - \alpha_c \frac{\partial \ell_c}{\partial \theta_c}$, where: $\ell_c = \frac{1}{|B_c|} \sum_{(s,a,s') \sim B_c} \frac{\pi(a|s)}{\widetilde{\mu}(a|s)} \left( R^\lambda(s,a) - \widehat{V}(s) \right)^2$
15:      **end for**
16:      Sample a mini-batch $B_a \sim \mathcal{D}'$ according to $\widehat{d}_\gamma$
17:      $z_w \leftarrow \frac{1}{|B_a|} \sum_{s \sim B_a} w(s)$
18:      $Q^\pi(s,a) \leftarrow \mathbb{1}\{(s,a) \in \mathcal{SA}_\mu\} R^\lambda(s,a)$
19:      $\theta \leftarrow \theta - \frac{\alpha}{|B_a|} \sum_{s \sim B_a} \frac{w(s)}{z_w} \rho(s,a) \frac{\partial \log \pi(a|s)}{\partial \theta} Q^\pi(s,a)$
20: **end for**

---

## B. Proof of Theorem 3

We first state and prove an abstract result. Suppose we have a function $f : \mathbb{R}^d \to \mathbb{R}$ which is differentiable, $G$-Lipschitz and $L$-smooth, and $f$ attains a finite minimum value $f^* := \min_{x \in \mathbb{R}^d} f(x)$. Suppose we have access to a noisy gradient oracle which returns a vector $\zeta(x) \in \mathbb{R}^d$ given a query point $x$. We say that the vector is $\sigma, B$-accurate for parameter $\sigma, B \geq 0$ if for all $x \in \mathbb{R}^d$, the quantity $\delta(x) := \zeta(x) - \nabla f(x)$ satisfies

$$\|\mathbb{E}\left[\delta(x) \mid x\right]\| \leq B \quad \text{and} \quad \mathbb{E}\left[\|\delta(x)\|^2 \mid x\right] \leq 2(\sigma^2 + B^2). \tag{6}$$

Notice that the expectations above are only with respect to any randomness in the oracle, while holding the query point fixed. Suppose we run the stochastic gradient descent algorithm using the oracle responses, that is we update $x_{k+1} = x_k - \eta \zeta(x_k)$. While several results for the convergence of stochastic gradient descent to a stationary point of a smooth, non-convex function are well-known, we could not find a result for the biased oracle assumed here and hence we provide a result from first principles. We have the following guarantee on the convergence of the sequence $x_k$ to an approximate stationary point of $f$.

**Theorem 4.** *Suppose $f$ is differentiable and $L$-smooth, and the approximate gradient oracle satisfies the conditions* (6) *with parameters $(\sigma_k, B_k)$ at iteration $k$. Then stochastic gradient descent with the oracle, with an initial solution $x_1$ and stepsize $\eta = 1/L$ satisfies after $K$ iterations:*

$$\frac{1}{K} \sum_{k=1}^K \mathbb{E}[\|\nabla f(x_k)\|^2] \leq \frac{2}{K}(f(x_1) - f^*) + \frac{2}{LK} \sum_{k=1}^K (\sigma_k^2 + B_k^2).$$

---

[3] $\widehat{d}_\gamma = \frac{1}{\sum_{t=0}^T \gamma^t} \sum_{t=0}^T \gamma^t \widehat{d}_t(s)$, where $\widehat{d}_t(s)$ is the empirical state distribution at time step $t$ in dataset $\mathcal{D}'$

*Proof.* Since $f$ is $L$-smooth, we have

$$f(x_{k+1}) \le f(x_k) + \langle \nabla f(x_k), x_{k+1} - x_k \rangle + \frac{L}{2} \|x_{k+1} - x_k\|^2$$

$$= f(x_k) - \eta \langle \nabla f(x_k), \zeta(x_k) \rangle + \frac{L\eta^2}{2} \|\zeta(x_k)\|^2$$

$$= f(x_k) - \eta \langle \nabla f(x_k), \delta(x_k) + \nabla f(x_k) \rangle + \frac{L\eta^2}{2} \|\delta(x_k) + \nabla f(x_k)\|$$

$$= f(x_k) + \|\nabla f(x_k)\|^2 \left( \frac{L\eta^2}{2} - \eta \right) - (\eta - L\eta^2) \langle \nabla f(x_k), \delta(x_k) \rangle + \frac{L\eta^2}{2} \|\delta(x_k)\|^2 .$$

Here the first equality follows from our update rule while the remaining simply use the definition of $\delta$ along with algebraic manipulations. Now taking expectations of both sides, we obtain

$$\mathbb{E}[f(x_{k+1})] \le \mathbb{E}[f(x_k)] + \mathbb{E}[\|\nabla f(x_k)\|^2] \left( \frac{L\eta^2}{2} - \eta \right) + (\eta - L\eta^2)GB_k + L\eta^2(\sigma_k^2 + B_k^2),$$

where we have invoked the properties of the oracle to bound the last two terms. Summing over iterations $k = 1, 2, \ldots, K$, we obtain

$$\mathbb{E}[f(x_{k+1})] \le f(x_1) + \left( \frac{L\eta^2}{2} - \eta \right) \sum_{k=1}^{K} \mathbb{E}[\|\nabla f(x_k)\|^2] + \eta \sum_{k=1}^{K} (GB_k(1 - L\eta) + L\eta(\sigma_k^2 + B_k^2)).$$

Rearranging terms, and using that $f(x_{K+1}) \ge f^*$, we obtain

$$\frac{1}{K} \sum_{k=1}^{K} \mathbb{E}[\|\nabla f(x_k)\|^2] \le \frac{f(x_1) - f(x^*)}{K(\eta - L\eta^2/2)} + \frac{\eta \sum_{k=1}^{K} (GB_k(1 - L\eta) + L\eta(\sigma_k^2 + B_k^2))}{K(\eta - L\eta^2/2)}.$$

Now using the choice $\eta = 1/L$ and simplifying, we obtain the statement of the theorem. $\qquad\square$

The theorem tells us that if we pick an iterate uniformly at random from $x_1, \ldots, x_K$, then it is an approximate stationary point in expectation, up to an accuracy which is determined by the bias and variance of the stochastic gradient oracle.

Given this abstract result, we can now prove Theorem 3 by instantiating the errors in the gradient oracle as a function of our assumptions.

**Proof of Theorem 3**    We now instantiate the result and assumptions for the case of the off-policy policy gradient method. First, note that the algorithm is stochastic gradient ascent for maximizing the expected return $J(\theta) := R^{\pi_\theta}$. Thus we can apply Theorem 4 with $f = -J$, so that $f(x_1) - f^* \le V_{\max}$ where $V_{\max}$ is an upper bound on the value of any policy in the MDP. $f$ attains a finite minimum value since the expected return has a finite maximum value. We focus on quantifying the bias $B$ in terms of errors in the critic and propensity score computations first. We first introduce some additional notation. Suppose $w_\theta(s)$ and $Q^\theta(s, a)$ are the true propensity (in terms of state distributions, relative to $\mu$) and $Q$-value functions for a policy $\pi_\theta$. Let $g_\theta(s, a) = \frac{\partial \log \pi_\theta(a|s)}{\partial \theta}$. Suppose we are given estimators $\hat{w}$ and $\widehat{Q}$ for $w_\theta$ and $Q^\theta$ respectively. Then our estimated and true off-policy policy gradients can be written as:

$$\nabla_\theta J(\theta) = \mathbb{E}_\nu[w \rho g_\theta Q^\pi] \quad \text{and} \quad \zeta(\theta) = \hat{w} \rho g_\theta \widehat{Q}.$$

Now the bias can be bounded as

$$\|\mathbb{E}[\zeta(\theta) - \nabla J(\theta)|\theta]\| = \left\| \mathbb{E}_\nu[w \rho g_\theta Q^\theta - \hat{w} \rho g_\theta \widehat{Q}] \right\|$$

$$\le \left\| \mathbb{E}_\nu[(w - \hat{w}) \rho g_\theta Q^\theta] \right\| + \left\| \mathbb{E}_\nu[\hat{w} \rho g_\theta (Q^\theta - \widehat{Q})] \right\|.$$

How we simplify further depends on the assumptions we make on the errors in $\hat{w}$ and $\widehat{Q}$. As a natural assumption, suppose that the relative errors are bounded in MSE, that is $\mathbb{E}_\nu(w(s) - \hat{w}(s))^2 \leq \varepsilon_w^2$ and $\mathbb{E}_\nu \left( Q^\theta(s,a) - \widehat{Q}(s,a) \right)^2 \leq \varepsilon_Q^2$. Then by Cauchy-Shwartz inequality, we can simplify the above bias term as

$$\| \mathbb{E}[\zeta(\theta) - \nabla J(\theta)|\theta] \| \leq \varepsilon_w \left\| \sqrt{\mathbb{E}_\nu[\rho g_\theta Q^\theta]^2} \right\| + \varepsilon_Q \left\| \sqrt{\mathbb{E}_\nu[\rho g_\theta \hat{w}]^2} \right\|,$$

where the operations of squaring and square root are applied elementwise to the vector $g_\theta$. By Assumption 2 we have $Q^\theta \leq V_{\max}$, $\hat{w}(s) \leq w_{\max}$ for all $s, a$, and

$$\| \rho(s,a) g_\theta(s,a) \| = \left\| \frac{1}{\nu(a|s)} \frac{\partial \pi_\theta(a|s)}{\partial \theta} \right\| \leq \frac{G_{\max}}{\nu_{\min}}$$

Then the bound on the bias further simplifies to

$$\| \mathbb{E}[\zeta(\theta) - \nabla J(\theta)|\theta] \| \leq \varepsilon_w G_{\max} V_{\max}/\nu_{\min} + \varepsilon_Q G_{\max} w_{\max}/\nu_{\min}$$

Similarly, for the variance we have

$$\mathbb{E}[\|\zeta(\theta) - \nabla J(\theta)\|^2 | \theta] \leq 2\mathbb{E}_\nu[\|(\hat{w} - w)\rho g_\theta Q^\theta\|^2] + 2\mathbb{E}_\nu[\|\hat{w}\rho g_\theta(\widehat{Q} - Q^\theta)\|^2]$$
$$\leq 2\varepsilon_w^2 G_{\max}^2 V_{\max}^2/\nu_{\min}^2 + 2\varepsilon_Q^2 w_{\max}^2 G_{\max}^2/\nu_{\min}^2.$$

Hence, the RHS of Theorem 4 simplifies to

$$\frac{2V_{\max}}{K} + O\left( \frac{\sum_{k=1}^T \varepsilon_{w,k}^2 G_{\max}^2 V_{\max}^2/\nu_{\min}^2 + \varepsilon_{Q,k}^2 w_{\max}^2 G_{\max}^2/\nu_{\min}^2}{K} \right),$$

where $\varepsilon_{w,k}$ and $\varepsilon_{Q,k}$ are the error parameters in the propensity scores and critic at the $k_{th}$ iteration of our algorithm. Since we update these quantities online along with the policy parameters, we expect $\varepsilon_{w,k}$ and $\varepsilon_{Q,k}$ to decrease as $k$ increases. That is, assuming that $\nu$ satisfies the coverage assumptions with finite upper bounds on the propensities and the policy class is Lipschitz continuous in its parameters, the scheme converges to an approximate stationary point given estimators $\hat{w}$ and $\widehat{Q}$ with a small average MSE across the iterations under $\nu$.

## C. Details for Experiments

In this section we will show some important details and hyper-parameter settings of our algorithm in experiments. We use three separate neural networks, one for each of actor, critic, and the state distribution ratio model $w$. We use the Adam optimizer for all of them. We also use a entropy regularization for the actor. We warm start the actor by maximizing the log-likelihood of actor on the collected dataset. For critic, we use the same critic algorithm as we used in Algorithm 1 except that there is no importance sampling ratio (as it is on-policy for the warm start). For the warm start of w, we just fit the w for several iterations using the warm start policy found for the actor. Warm start uses the same learning rates as normal training. For critic and $w$, we also keep the state of optimizer to be the same when we start normal training.

In the table below we show some hyper-parameters setting we used in both domain:

We also follow the details in Algorithm 1 and Algorithm 2 of Liu et al. (2018a) to learn $w$. We scale the inputs to $w$ so that the whole off-policy dataset has zero mean and standard deviation of 1 along each dimension in state space. We use the RBF kernel to compute the loss function for $w$. For the CartPole simulator, the kernel bandwidth is set to be the median of state distance. If computing this median state distance over the whole off-policy dataset is computationally too expensive, it can be approximated using a mini-batch. In the HIV domain the bandwidth is set to be 1. When we compute the loss of $w$, we need to sample two mini-batch independently to get an unbiased estimates of the quadratic loss. The loss in each pair of mini-batch is normalized by the sum of kernel matrix elements computed from them.

| Hyper-parameters | cart pole | HIV |
|---|---|---|
| $\gamma$ | 1.0 | 0.98 |
| $\lambda$ | 0. | 0. |
| entropy coefficient | 0.01 | 0.03 |
| learning rate (actor) | 1e-3 | 5e-6 |
| learning rate (critic) | 1e-3 | 1e-3 |
| learning rate (w) | 1e-3 | 3e-4 |
| batch size (actor) | 5000 | 5000 |
| batch size (critic) | 5000 | 5000 |
| batch size (w) | 200 | 200 |
| number of iterations (critic) | 10 | 10 |
| number of iterations (w) | 50 | 50 |
| weight decay (w) | 1e-5 | 1e-5 |
| behavior cloning number of iterations | 2000 | 2000 |
| warm start number of iterations (crtic) | 500 | 2500 |
| warm start number of iterations (w) | 500 | 2500 |

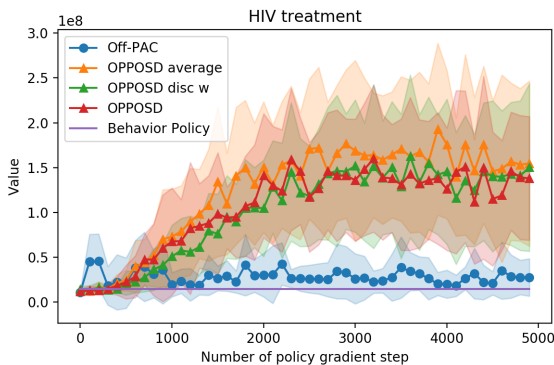

*Figure 4.* Episodic scores over length 200 episodes in HIV treatment simulator.

## C.1. Choice of Algorithm with Discounted Reward

In discounted reward settings, the state distribution is also defined with respect to the discount factor $\gamma$, and Liu et al. (2018a) introduce an algorithm to learn state distribution ratio in this setting. However, we notice that in on-policy policy learning cases, though the policy gradient theorem (Sutton et al., 2000) requires samples from the stationary state distribution defined using the discount factor, it is common to directly use the collected samples to compute policy gradient without re-sampling/re-weighting $(s, a, r, s')$'s according to the discounted stationary distribution. This might be driven by sample efficiency concerns, as samples at later time-steps in the discounted stationay distribution will receive exponentially small probability, meaning they are not leveraged as effectively by the algorithm. Given this, we compare three different variants of our algorithm in HIV experiment with discounted reward. The first (OPPOSD average) variant uses the algorithm for the average reward setting, but evaluates its discounted reward. The second learns the state distribution ratio $w(s)$ in the discounted case (Algorithm 2 in (Liu et al., 2018a)), but still samples from the undiscounted distribution to compute the gradient (OPPOSD disc $w$). The third learns the state distribution ratio $w(s)$ in the discounted case and also re-samples the samples according to $d^\pi(s) = \lim_{T \to \infty} \frac{1}{\sum_{t=0}^{T} \gamma^t} \sum_{t=0}^{T} \gamma^t d_t^\pi(s)$ (OPPOSD). In the main body of paper, we select the third one as it is the most natural way from the definition of problem and policy gradient theorem. Results of these three methods are demonstrated in Figure 4 and they do not have significant differences in this experiment.