# OpenReview forum: "Off-Policy Policy Gradient with State Distribution Correction"
_ICML.cc/2019/Workshop/RL4RealLife — RL4RealLife 2019_

### Official Review · AnonReviewer2 · 2019-05-15
**Nice preliminary results; would be nice to have more detailed results**

**Rating:** 4
**Confidence:** 4

**Review:**

The authors propose to perform policy gradient on off-policy data by appropriately weighting the off-policy experience by stationary distribution ratios, which themselves are estimated using recently-proposed TD-learning methods.  The authors show that this method is equivalent to the policy gradient theorem w.r.t. off-policy data.  The authors provide a theoretical and empirical comparison with Off-PAC.  Results on Cartpole and an HIV simulator show strong benefits over Off-PAC.

Comments:
-- The idea is nice and the method is simple.
-- The idea of weighting RL learning updates based on some propensity score has been proposed by others. See https://arxiv.org/abs/1507.01569 and https://arxiv.org/abs/1901.09455 . I would like to see more discussion about these previous algorithms.
-- The empirical results are impressive.
-- While the results are impressive, they are preliminary.  I would like to see comparisons to the previous propensity score methods above as well as standard off-policy RL algorithms (e.g., Q-learning) and standard on-policy algorithms (e.g. actor-critic).
-- I would also like to understand the benefits of the method better.  For example, the propensity-score estimation relies on a kernel.  How much does a good kernel benefit the overall method?

---

### Official Review · AnonReviewer1 · 2019-05-26
**Extension on prior work on policy evaluation using state distribution correction to off-policy actor critic**

**Rating:** 2
**Confidence:** 4

**Review:**

The work extends prior work (Liu et. al 2018) on Infinite-Horizon Off-Policy Estimation to the case of off-policy policy optimization with actor-critic. The authors compared the proposed method to baseline off-policy actor critic and demonstrated the effectiveness of state-distribution correction in multiple simulation environments.

The novelty of the work is limited. The idea of state-distribution correction under infinite horizon was covered in prior work. The augmented MDP with absorbing state and obtaining pessimistic policy evaluation or optimization is new, but the authors did not extend on its importance in the method or experiments.

The authors did not investigate how much the infinite horizon assumption affect the performance.

---

### Decision · Program_Chairs · 2019-05-28

Accept